# PC-DARTS: Partial Channel Connections for Memory-Efficient Architecture Search

**Yuhui Xu**[1]* **Lingxi Xie**[2] **Xiaopeng Zhang**[2] **Xin Chen**[3]
**Guo-Jun Qi**[4] **Qi Tian**[2(✉)] **Hongkai Xiong**[1]
[1]Shanghai Jiao Tong University  [2]Huawei Noah's Ark Lab
[3]Tongji University  [4]Futurewei Technologies
`yuhuixu@sjtu.edu.cn` `{198808xc,zxphistory}@gmail.com` `1410452@tongji.edu.cn`
`guojunq@gmail.com` `tian.qi1@huawei.com` `xionghongkai@sjtu.edu.cn`

## Abstract

Differentiable architecture search (DARTS) provided a fast solution in finding effective network architectures, but suffered from large memory and computing overheads in jointly training a super-network and searching for an optimal architecture. In this paper, we present a novel approach, namely, **Partially-Connected DARTS**, by sampling a small part of super-network to reduce the redundancy in exploring the network space, thereby performing a more efficient search without comprising the performance. In particular, we perform operation search in a subset of channels while bypassing the held out part in a shortcut. This strategy may suffer from an undesired inconsistency on selecting the edges of super-net caused by sampling different channels. We alleviate it using *edge normalization*, which adds a new set of edge-level parameters to reduce uncertainty in search. Thanks to the reduced memory cost, PC-DARTS can be trained with a larger batch size and, consequently, enjoys both faster speed and higher training stability. Experimental results demonstrate the effectiveness of the proposed method. Specifically, we achieve an error rate of $2.57\%$ on CIFAR10 with merely 0.1 GPU-days for architecture search, and a state-of-the-art top-1 error rate of $24.2\%$ on ImageNet (under the mobile setting) using 3.8 GPU-days for search. *Our code has been made available at* `https://github.com/yuhuixu1993/PC-DARTS`.

## 1 Introduction

Neural architecture search (NAS) emerged as an important branch of automatic machine learning (AutoML), and has been attracting increasing attentions from both academia and industry. The key methodology of NAS is to build a large space of network architectures, develop an efficient algorithm to explore the space, and discover the optimal structure under a combination of training data and constraints (*e.g.*, network size and latency). Different from early approaches that often incur large computation overheads (Zoph & Le, 2017; Zoph et al., 2018; Real et al., 2019), recent one-shot approaches (Pham et al., 2018; Liu et al., 2019) have reduced the search costs by orders of magnitudes, which advances its applications to many real-world problems. In particular, DARTS (Liu et al., 2019) converts the operation selection into weighting a fixed set of operations. This makes the entire framework differentiable to architecture hyper-parameters and thus the network search can be efficiently accomplished in an end-to-end fashion. Despite its sophisticated design, DARTS is still subject to a large yet redundant space of network architectures and thus suffers from heavy memory and computation overheads. This prevents the search process from using larger batch sizes for either speedup or higher stability. Prior work (Chen et al., 2019) proposed to reduce the search space, which leads to an approximation that may sacrifice the optimality of the discovered architecture.

---

* This work was done when Yuhui Xu and Xin Chen were interns at Huawei Noah's Ark Lab.
✉ Qi Tian is the corresponding author.
This work was supported in part by the National NSFC under Grant Nos. 61971285, 61425011, 61529101, 61622112, 61720106001, 61932022, and in part by the Program of Shanghai Academic Research Leader under Grant 17XD1401900. We thank Longhui Wei and Bowen Shi for instructive discussions.

In this paper, we present a simple yet effective approach named Partially-Connected DARTS (PC-DARTS) to reduce the burdens of memory and computation. The core idea is intuitive: instead of sending all channels into the block of operation selection, we randomly sample a subset of them in each step, while bypassing the rest directly in a shortcut. We assume the computation on this subset is a surrogate approximating that on all the channels. Besides the tremendous reduction in memory and computation costs, channel sampling brings another benefit – operation search is regularized and less likely to fall into local optima. However, PC-DARTS incurs a side effect, where the selection of channel connectivity would become unstable as different subsets of channels are sampled across iterations. Thus, we introduce *edge normalization* to stabilize the search for network connectivity by explicitly learning an extra set of edge-selection hyper-parameters. By sharing these hyper-parameters throughout the training process, the sought network architecture is insensitive to the sampled channels across iterations and thus is more stable.

Benefiting from the partial connection strategy, we are able to greatly increase the batch size. Specifically, as only $1/K$ of channels are randomly sampled for an operation selection, it reduces the memory burden by almost $K$ times. This allows us to use a $K$ times larger batch size during search, which not only accelerates the network search but also stabilizes the process particularly for large-scale datasets. Experiments on benchmark datasets demonstrate the effectiveness of PC-DARTS. Specifically, we achieve an error rate of 2.57% in less than 0.1 GPU-days (around 1.5 hours) on a single Tesla V100 GPU, surpassing the result of 2.76% reported by DARTS that required 1.0 GPU-day. Furthermore, PC-DARTS allows a direct search on ImageNet (while DARTS failed due to low stability), and sets the state-of-the-art record with a top-1 error of 24.2% (under the mobile setting) in only 3.8 GPU-days (11.5 hours on eight Tesla V100 GPUs).

## 2 RELATED WORK

Thanks to the rapid development of deep learning, significant gain in performance has been brought to a wide range of computer vision problems, most of which owed to manually desgined network architectures (Krizhevsky et al., 2012; Simonyan & Zisserman, 2015; He et al., 2016; Huang et al., 2017). Recently, a new research field named neural architecture search (NAS) has been attracting increasing attentions. The goal is to find automatic ways of designing neural architectures to replace conventional handcrafted ones. According to the heuristics to explore the large architecture space, existing NAS approaches can be roughly divided into three categories, namely, evolution-based approaches, reinforcement-learning-based approaches and one-shot approaches.

The first type of architecture search methods (Liu et al., 2018b; Xie & Yuille, 2017; Real et al., 2017; Elsken et al., 2019; Real et al., 2019; Miikkulainen et al., 2019) adopted evolutionary algorithms, which assumed the possibility of applying genetic operations to force a single architecture or a family evolve towards better performance. Among them, Liu *et al.* (Liu et al., 2018b) introduced a hierarchical representation for describing a network architecture, and Xie *et al.* (Xie & Yuille, 2017) decomposed each architecture into a representation of 'genes'. Real *et al.* (Real et al., 2019) proposed aging evolution which improved upon standard tournament selection, and surpassed the best manually designed architecture since then. Another line of heuristics turns to reinforcement learning (RL) (Zoph & Le, 2017; Baker et al., 2017; Zoph et al., 2018; Zhong et al., 2018; Liu et al., 2018a), which trained a meta-controller to guide the search process. Zoph *et al.* (Zoph & Le, 2017) first proposed using a controller-based recurrent neural network to generate hyper-parameters of neural networks. To reduce the computation cost, researchers started to search for blocks or cells (Zhong et al., 2018; Zoph et al., 2018) instead of the entire network, and consequently, managed to reduce the overall computational costs by a factor of 7. Other kinds of approximation, such as greedy search (Liu et al., 2018a), were also applied to further accelerate search. Nevertheless, the computation costs of these approaches, based on either evolution or RL, are still beyond acceptance.

In order to accomplish architecture search within a short period of time, researchers considered to reduce the costs of evaluating each searched candidate. Early efforts include sharing weights between searched and newly generated networks (Cai et al., 2018), and later these methods were generalized into a more elegant framework named one-shot architecture search (Brock et al., 2018; Cai et al., 2019; Liu et al., 2019; Pham et al., 2018; Xie et al., 2019), in which an over-parameterized network or super-network covering all candidate operations was trained only once, from which exponentially many sub-networks can be sampled. As typical examples, SMASH (Brock et al.,

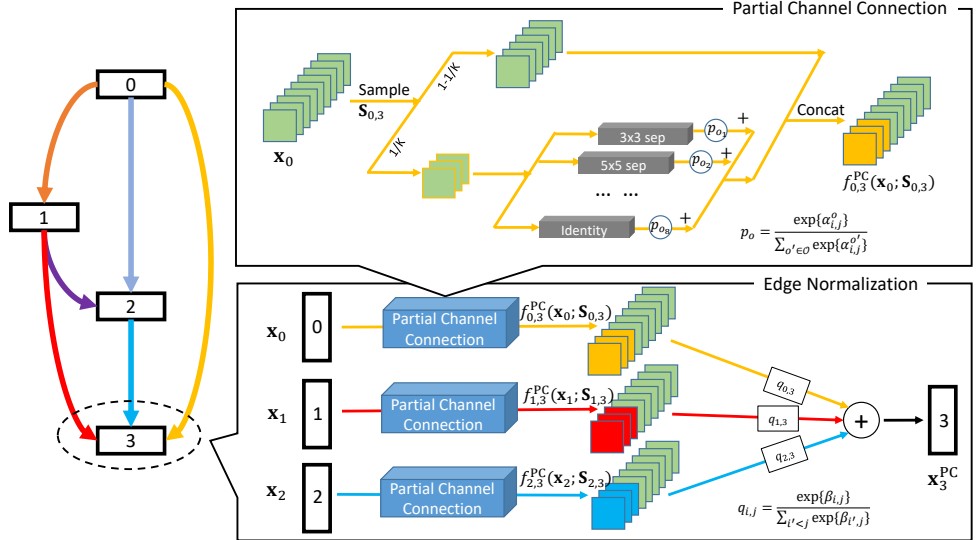

Figure 1: Illustration of the proposed approach (best viewed in color), partially-connected DARTS (PC-DARTS). As an example, we investigate how information is propagated to node #3, *i.e.*, $j = 3$. There are two sets of hyper-parameters during search, namely, $\{\alpha_{i,j}^o\}$ and $\{\beta_{i,j}\}$, where $0 \leqslant i < j$ and $o \in \mathcal{O}$. To determine $\{\alpha_{i,j}^o\}$, we only sample a subset, $1/K$, of channels and connect them to the next stage, so that the memory consumption is reduced by $K$ times. To minimize the uncertainty incurred by sampling, we add $\{\beta_{i,j}\}$ as *extra* edge-level parameters.

2018) trained the over-parameterized network by a HyperNet (Ha et al., 2017), and ENAS (Pham et al., 2018) shared parameters among child models to avoid retraining each candidate from scratch.

This paper is based on DARTS (Liu et al., 2018b), which introduced a differentiable framework for architecture search, and thus combine the search and evaluation stages into one. A super-network is optimized during the search stage, after which the strongest sub-network is preserved and then retrained. Despite its simplicity, researchers detected some of its drawbacks, such as instability (Li & Talwalkar, 2019; Sciuto et al., 2019), which led to a few improved approaches beyond DARTS (Cai et al., 2019; Chen et al., 2019; Mei et al., 2020). In particular, ProxylessNAS (Cai et al., 2019) was the first method that searched directly on ImageNet, and P-DARTS (Chen et al., 2019) designed a progressive search stage to bridge the depth gap between the super-network and the sub-network.

## 3 THE PROPOSED APPROACH

### 3.1 PRELIMINARIES: DIFFERENTIABLE ARCHITECTURE SEARCH (DARTS)

We first review the baseline DARTS (Liu et al., 2019), and define the notations for the discussion later. Mathematically, DARTS decomposes the searched network into a number ($L$) of cells. Each cell is represented as a directed acyclic graph (DAG) with $N$ nodes, where each node defines a network layer. There is a pre-defined space of operations denoted by $\mathcal{O}$, in which each element, $o(\cdot)$, is a fixed operation (*e.g.*, identity connection, and $3 \times 3$ convolution) performed at a network layer. Within a cell, the goal is to choose one operation from $\mathcal{O}$ to connect each pair of nodes. Let a pair of nodes be $(i, j)$, where $0 \leqslant i < j \leqslant N - 1$, the core idea of DARTS is to formulate the information propagated from $i$ to $j$ as a weighted sum over $|\mathcal{O}|$ operations, namely, $f_{i,j}(\mathbf{x}_i) = \sum_{o \in \mathcal{O}} \frac{\exp\{\alpha_{i,j}^o\}}{\sum_{o' \in \mathcal{O}} \exp\{\alpha_{i,j}^{o'}\}} \cdot o(\mathbf{x}_i)$, where $\mathbf{x}_i$ is the output of the $i$-th node, and $\alpha_{i,j}^o$ is a hyper-parameter for weighting operation $o(\mathbf{x}_i)$. The output of a node is the sum of all input flows, *i.e.*, $\mathbf{x}_j = \sum_{i<j} f_{i,j}(\mathbf{x}_i)$, and the output of the entire cell is formed by concatenating the output of nodes $\mathbf{x}_2$–$\mathbf{x}_{N-1}$, *i.e.*, $\text{concat}(\mathbf{x}_2, \mathbf{x}_3, \ldots, \mathbf{x}_{N-1})$. Note that the first two nodes, $\mathbf{x}_0$ and $\mathbf{x}_1$, are input nodes to a cell, which are fixed during architecture search.

This design makes the entire framework differentiable to both layer weights and hyper-parameters $\alpha_{i,j}^o$, so that it is possible to perform architecture search in an end-to-end fashion. After the search process is finished, on each edge $(i,j)$, the operation $o$ with the largest $\alpha_{i,j}^o$ value is preserved, and each node $j$ is connected to two precedents $i < j$ with the largest $\alpha_{i,j}^o$ preserved.

## 3.2 Partial Channel Connections

A drawback of DARTS lies in memory inefficiency. In the main part of the searched architecture, $|\mathcal{O}|$ operations and the corresponding outputs need to be stored at each node (*i.e.*, each network layer), leading to $|\mathcal{O}|\times$ memory to use. To fit into a GPU, one must reduce the batch size during search, which inevitably slows down search speed, and may deteriorate search stability and accuracy.

An alternative solution to memory efficiency is the **partial channel connection** as depicted in Figure 1. Take the connection from $\mathbf{x}_i$ to $\mathbf{x}_j$ for example. This involves defining a channel sampling mask $\mathbf{S}_{i,j}$, which assigns 1 to selected channels and 0 to masked ones. The selected channels are sent into mixed computation of $|\mathcal{O}|$ operations, while the masked ones bypass these operations, *i.e.*, they are directly copied to the output,

$$f_{i,j}^{\mathrm{PC}}(\mathbf{x}_i; \mathbf{S}_{i,j}) = \sum_{o \in \mathcal{O}} \frac{\exp\left\{\alpha_{i,j}^o\right\}}{\sum_{o' \in \mathcal{O}} \exp\left\{\alpha_{i,j}^{o'}\right\}} \cdot o(\mathbf{S}_{i,j} * \mathbf{x}_i) + (1 - \mathbf{S}_{i,j}) * \mathbf{x}_i. \tag{1}$$

where, $\mathbf{S}_{i,j} * \mathbf{x}_i$ and $(1 - \mathbf{S}_{i,j}) * \mathbf{x}_i$ denote the selected and masked channels, respectively. In practice, we set the proportion of selected channels to $1/K$ by regarding $K$ as a hyper-parameter. By varying $K$, we could trade off between architecture search accuracy (smaller $K$) and efficiency (larger $K$) to strike a balance (See Section 4.4.1 for more details).

A direct benefit brought by the partial channel connection is that the memory overhead of computing $f_{i,j}^{\mathrm{PC}}(\mathbf{x}_i; \mathbf{S}_{i,j})$ is reduced by $K$ times. This allows us to use a larger batch size for architecture search. There are twofold benefits. First, the computing cost could be reduced by $K$ times during the architecture search. Moreover, the larger batch size implies the possibility of sampling more training data during each iteration. This is particularly important for the stability of architecture search. In most cases, the advantage of one operation over another is not significant, unless more training data are involved in a mini-batch to reduce the uncertainty in updating the parameters of network weights and architectures.

## 3.3 Edge Normalization

Let us look into the impact of sampling channels on neural architecture search. There are both positive and negative effects. On the **upside**, by feeding a small subset of channels for operation mixture while bypassing the remainder, we make it less biased in selecting operations. In other words, for edge $(i,j)$, given an input $\mathbf{x}_i$, the difference from using two sets of hyper-parameters $\left\{\alpha_{i,j}^o\right\}$ and $\left\{\alpha_{i,j}'^o\right\}$ is largely reduced, because only a small part $(1/K)$ of input channels would go through the operation mixture while the remaining channels are left intact. This regularizes the preference of a weight-free operation (*e.g.*, *skip-connect*, *max-pooling*, *etc.*) over a weight-equipped one (*e.g.*, various kinds of *convolution*) in $\mathcal{O}$. In the early stage, the search algorithm often prefers weight-free operations, because they do not have weights to train and thus produce more consistent outputs, *i.e.*, $o(\mathbf{x}_i)$. In contrast, the weight-equipped ones, before their weights are well optimized, would propagate inconsistent information across iterations. Consequently, weight-free operations often accumulate larger weights (namely $\alpha_{i,j}^o$) at the beginning, and this makes it difficult for the weight-equipped operations to beat them even after they have been well trained thereafter. *This phenomenon is especially significant when the proxy dataset (on which architecture search is performed) is difficult, and this could prevent DARTS from performing satisfactory architecture search on ImageNet.* In experiments, we will show that PC-DARTS, with partial channel connections, produces more stable and superior performance on ImageNet.

On the **downside**, in a cell, each output node $\mathbf{x}_j$ needs to pick up two input nodes from its precedents $\{\mathbf{x}_0, \mathbf{x}_1, \ldots, \mathbf{x}_{j-1}\}$, which are weighted by $\max_o \alpha_{0,j}^o, \max_o \alpha_{1,j}^o, \ldots, \max_o \alpha_{j-1,j}^o$, respectively, following the original DARTS. However, these architecture parameters are optimized by randomly sampled channels across iterations, and thus the optimal connectivity determined by them could be unstable as the sampled channels change over time. This could cause undesired fluctuation in

the resultant network architecture. To mitigate this problem, we introduce edge normalization that weighs on each edge $(i, j)$ explicitly, denoted by $\beta_{i,j}$, so that the computation of $\mathbf{x}_j$ becomes:

$$\mathbf{x}_j^{\text{PC}} = \sum_{i<j} \frac{\exp\{\beta_{i,j}\}}{\sum_{i'<j} \exp\{\beta_{i',j}\}} \cdot f_{i,j}(\mathbf{x}_i). \tag{2}$$

Specifically, after the architecture search is done, the connectivity of edge $(i, j)$ is determined by both $\{\alpha_{i,j}^o\}$ and $\beta_{i,j}$, for which we multiply the normalized coefficients together, *i.e.*, multiplying $\frac{\exp\{\beta_{i,j}\}}{\sum_{i'<j} \exp\{\beta_{i',j}\}}$ by $\frac{\exp\{\alpha_{i,j}^o\}}{\sum_{o' \in \mathcal{O}} \exp\{\alpha_{i,j}^{o'}\}}$. Then the edges are selected by finding the large edge weights as in DARTS. Since $\beta_{i,j}$ are shared through the training process, the learned network architecture is insensitive to the sampled channels across iterations, making the architecture search more stable. In Section 4.4.2, we will show that edge normalization is also effective over the original DARTS. Finally, the extra computation overhead required for edge normalization is negligible.

### 3.4 DISCUSSIONS AND RELATIONSHIP TO PRIOR WORK

First of all, there are two major contributions of our approach, namely, channel sampling and edge normalization. Channel sampling, as the key technique in this work, has not been studied in NAS for reducing computational overhead (other regularization methods like Dropout (Srivastava et al., 2014) and DropPath (Larsson et al., 2017) cannot achieve the same efficiency, in both time and memory, as channel sampling). It accelerates and regularizes search and, with the help of edge normalization, improves search stability. Note that both search speed and stability are very important for a search algorithm. Combining channel sampling and edge normalization, we obtain the best accuracy on ImageNet (based on the DARTS search space), and the direct search cost on ImageNet (3.8 GPU-days) is the lowest known. Moreover, these two components are easily transplanted to other search algorithms to improve search accuracy and speed, *e.g.*, edge normalization boosts the accuracy and speed of the original DARTS methods.

Other researchers also tried to alleviate the large memory consumption of DARTS. Among prior efforts, ProxylessNAS (Cai et al., 2019) binarized the multinomial distribution $\alpha_{i,j}^o$ and samples two paths at each time, which significantly reduced memory cost and enabled direct search on ImageNet. PARSEC (Casale et al., 2019) also proposed a sampling-based optimization method to learn a probability distribution. Our solution, by preserving all operations for architecture search, achieves a higher accuracy in particular on challenging datasets like ImageNet ($+0.7\%$ over ProxylessNAS and $+1.8\%$ over PARSEC). Another practical method towards memory efficiency is Progressive-DARTS (Chen et al., 2019), which eliminated a subset of operators in order to provide sufficient memory for deeper architecture search. In comparison, our approach preserves all operators and instead performs sub-sampling on the channel dimension. This strategy works better in particular on large-scale datasets like ImageNet.

## 4 EXPERIMENTS

### 4.1 DATASETS AND IMPLEMENTATION DETAILS

We perform experiments on CIFAR10 and ImageNet, two most popular datasets for evaluating neural architecture search. CIFAR10 (Krizhevsky & Hinton, 2009) consists of 60K images, all of which are of a spatial resolution of $32 \times 32$. These images are equally distributed over 10 classes, with 50K training and 10K testing images. ImageNet (Deng et al., 2009) contains 1,000 object categories, and 1.3M training images and 50K validation images, all of which are high-resolution and roughly equally distributed over all classes. Following the conventions (Zoph et al., 2018; Liu et al., 2019), we apply the *mobile setting* where the input image size is fixed to be $224 \times 224$ and the number of multi-add operations does not exceed 600M in the testing stage.

Following DARTS (Liu et al., 2019) as well as conventional architecture search approaches, we use an individual stage for architecture search, and after the optimal architecture is obtained, we conduct another training process from scratch. In the search stage, the goal is to determine the best sets of hyper-parameters, namely $\{\alpha_{i,j}^o\}$ and $\{\beta_{i,j}\}$ for each edge $(i, j)$. To this end, the trainnig set is partitioned into two parts, with the first part used for optimizing network parameters, *e.g.*,

Table 1: Comparison with state-of-the-art network architectures on CIFAR10.

| Architecture | Test Err. (%) | Params (M) | Search Cost (GPU-days) | Search Method |
|---|---|---|---|---|
| DenseNet-BC (Huang et al., 2017) | 3.46 | 25.6 | - | manual |
| NASNet-A + cutout (Zoph et al., 2018) | 2.65 | 3.3 | 1800 | RL |
| AmoebaNet-B + cutout (Real et al., 2019) | 2.55±0.05 | 2.8 | 3150 | evolution |
| Hireachical Evolution (Liu et al., 2018b) | 3.75±0.12 | 15.7 | 300 | evolution |
| PNAS (Liu et al., 2018a) | 3.41±0.09 | 3.2 | 225 | SMBO |
| ENAS + cutout (Pham et al., 2018) | 2.89 | 4.6 | 0.5 | RL |
| NAONet-WS (Luo et al., 2018) | 3.53 | 3.1 | 0.4 | NAO |
| DARTS (1st order) + cutout (Liu et al., 2019) | 3.00±0.14 | 3.3 | 0.4 | gradient-based |
| DARTS (2nd order) + cutout (Liu et al., 2019) | 2.76±0.09 | 3.3 | 1 | gradient-based |
| SNAS (moderate) + cutout (Xie et al., 2019) | 2.85±0.02 | 2.8 | 1.5 | gradient-based |
| ProxylessNAS + cutout (Cai et al., 2019) | 2.08 | - | 4.0 | gradient-based |
| P-DARTS + cutout (Chen et al., 2019) | 2.50 | 3.4 | 0.3 | gradient-based |
| BayesNAS + cutout (Zhou et al., 2019) | 2.81±0.04 | 3.4 | 0.2 | gradient-based |
| PC-DARTS + cutout | 2.57±0.07[‡] | 3.6 | **0.1**[†] | gradient-based |

[†] Recorded on a single GTX 1080Ti. It can be shortened into 0.06 GPU-days if Tesla V100 is used.

[‡] We ran PC-DARTS 5 times and used standalone validation to pick the best from the 5 runs. This process was done by using 45K out of 50K training images for training, and the remaining 5K images for validation. The best one in validation was used for testing, which reported a test error of 2.57%.

convolutional weights, and the second part used for optimizing hyper-parameters. The entire search stage is accomplished in an end-to-end manner. For fair comparison, the operation space $\mathcal{O}$ remains the same as the convention, which contains 8 choices, *i.e.*, 3×3 and 5×5 *separable convolution*, 3×3 and 5×5 *dilated separable convolution*, 3×3 *max-pooling*, 3×3 *average-pooling*, *skip-connect* (*a.k.a.*, *identity*), and *zero* (*a.k.a.*, *none*).

We propose an alternative and more efficient implementation for partial channel connections. For edge $(i, j)$, we do not perform channel sampling at each time of computing $o(\mathbf{x}_i)$, but instead choose the first $1/K$ channels of $\mathbf{x}_i$ for operation mixture directly. To compensate, after $\mathbf{x}_j$ is obtained, we shuffle its channels before using it for further computations. This is the same implementation used in ShuffleNet (Zhang et al., 2018), which is more GPU-friendly and thus runs faster.

## 4.2 RESULTS ON CIFAR10

In the search scenario, the over-parameterized network is constructed by stacking 8 cells (6 normal cells and 2 reduction cells), and each cell consists of $N = 6$ nodes. We train the network for 50 epochs, with the initial number of channels being 16. The 50K training set of CIFAR10 is split into two subsets with equal size, with one subset used for training network weights and the other used for architecture hyper-parameters.

We set $K = 4$ for CIFAR10, *i.e.*, only $1/4$ features are sampled on each edge, so that the batch size during search is increased from 64 to 256. Besides, following (Chen et al., 2019), we freeze the hyper-parameters, $\left\{\alpha_{i,j}^o\right\}$ and $\{\beta_{i,j}\}$, and only allow the network parameters to be tuned in the first 15 epochs. This process, called warm-up, is to alleviate the drawback of the parameterized operations. The total memory cost is less than 12GB so that we can train it on most modern GPUs. The network weights are optimized by momentum SGD, with an initial learning rate of 0.1 (annealed down to zero following a cosine schedule without restart), a momentum of 0.9, and a weight decay of $3 \times 10^{-4}$. We use an Adam optimizer (Kingma & Ba, 2015) for $\left\{\alpha_{i,j}^o\right\}$ and $\{\beta_{i,j}\}$, with a fixed learning rate of $6 \times 10^{-4}$, a momentum of $(0.5, 0.999)$ and a weight decay of $10^{-3}$. Owing to the increased batch size, the entire search process only requires 3 hours on a GTX 1080Ti GPU, or 1.5 hours on a Tesla V100 GPU, which is almost $4\times$ faster than the original first-order DARTS.

The evaluation stage simply follows that of DARTS. The network is composed of 20 cells (18 normal cells and 2 reduction cells), and each type of cells share the same architecture. The initial number of channels is 36. The entire 50K training set is used, and the network is trained from scratch for

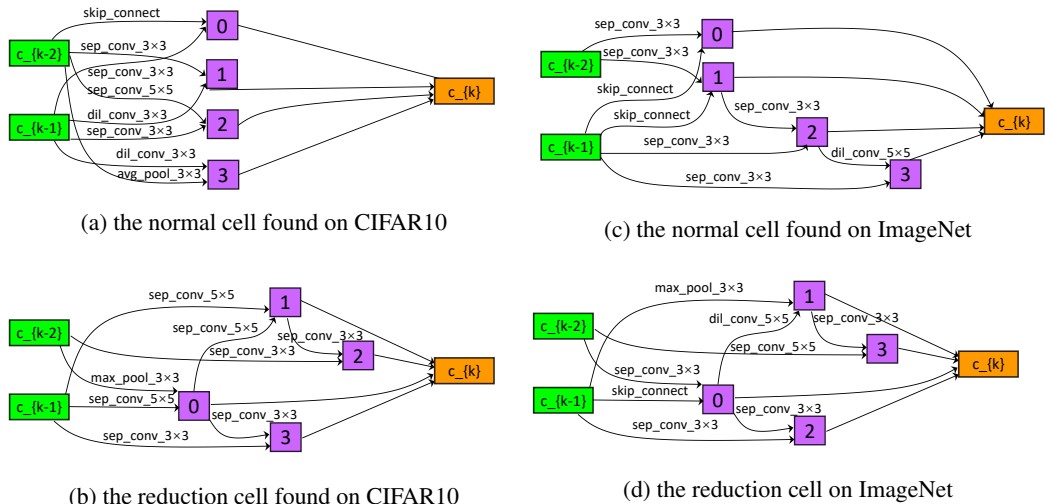

Figure 2: Cells found on CIFAR10 and ImageNet. Searching on ImageNet makes the normal cell more complex (deeper), although the reduction cell is very similar to that found on CIFAR10.

600 epochs using a batch size of 128. We use the SGD optimizer with an initial learning rate of 0.025 (annealed down to zero following a cosine schedule without restart), a momentum of 0.9, a weight decay of $3 \times 10^{-4}$ and a norm gradient clipping at 5. Drop-path with a rate of 0.3 as well as cutout (DeVries & Taylor, 2017) is also used for regularization. We visualize the searched normal and reduction cells in the left-hand side of Figure 2.

Results and comparison to recent approaches are summarized in Table 1. In merely 0.1 GPU-days, PC-DARTS achieve an error rate of 2.57%, with both search time and accuracy surpassing the baseline, DARTS, significantly. To the best of our knowledge, our approach is the fastest one that achieves an error rate of less than 3%. Our number ranks among the top of recent architecture search results. ProxylessNAS used a different protocol to achieve an error rate of 2.08%, and also reported a much longer time for architecture search. P-DARTS (Chen et al., 2019) slightly outperforms our approach by searching over a deeper architecture, which we can integrate our approach into P-DARTS to accelerate it as well as improve its performance (consistent accuracy gain is obtained).

### 4.3 RESULTS ON IMAGENET

We slightly modify the network architecture used on CIFAR10 to fit ImageNet. The over-parameterized network starts with three convolution layers of stride 2 to reduce the input image resolution from $224 \times 224$ to $28 \times 28$. 8 cells (6 normal cells and 2 reduction cells) are stacked beyond this point, and each cell consists of $N = 6$ nodes. To reduce search time, we randomly sample two subsets from the 1.3M training set of ImageNet, with 10% and 2.5% images, respectively. The former one is used for training network weights and the latter for updating hyper-parameters.

ImageNet is much more difficult than CIFAR10. To preserve more information, we use a sub-sampling rate of $1/2$, which doubles that used in CIFAR10. Still, a total of 50 epochs are trained and architecture hyper-parameters are frozen during the first 35 epochs. For network weights, we use a momentum SGD with an initial learning rate of 0.5 (annealed down to zero following a cosine schedule without restart), a momentum of 0.9, and a weight decay of $3 \times 10^{-5}$. For hyper-parameters, we use the Adam optimizer (Kingma & Ba, 2015) with a fixed learning rate of $6 \times 10^{-3}$, a momentum $(0.5, 0.999)$ and a weight decay of $10^{-3}$. We use eight Tesla V100 GPUs for search, and the total batch size is 1,024. The entire search process takes around 11.5 hours. We visualize the searched normal and reduction cells in the right-hand side of Figure 2.

The evaluation stage follows that of DARTS, which also starts with three convolution layers with a stride of 2 that reduce the input image resolution from $224 \times 224$ to $28 \times 28$. 14 cells (12 normal cells and 2 reduction cells) are stacked beyond this point, with the initial channel number being 48. The network is trained from scratch for 250 epochs using a batch size of 1,024. We use the SGD

Table 2: Comparison with state-of-the-art architectures on ImageNet (mobile setting).

| Architecture | Test Err. (%) | | Params | $\times+$ | Search Cost | Search Method |
|---|---|---|---|---|---|---|
| | top-1 | top-5 | (M) | (M) | (GPU-days) | |
| Inception-v1 (Szegedy et al., 2015) | 30.2 | 10.1 | 6.6 | 1448 | - | manual |
| MobileNet (Howard et al., 2017) | 29.4 | 10.5 | 4.2 | 569 | - | manual |
| ShuffleNet 2× (v1) (Zhang et al., 2018) | 26.4 | 10.2 | ∼5 | 524 | - | manual |
| ShuffleNet 2× (v2) (Ma et al., 2018) | 25.1 | - | ∼5 | 591 | - | manual |
| NASNet-A (Zoph et al., 2018) | 26.0 | 8.4 | 5.3 | 564 | 1800 | RL |
| AmoebaNet-C (Real et al., 2019) | 24.3 | 7.6 | 6.4 | 570 | 3150 | evolution |
| PNAS (Liu et al., 2018a) | 25.8 | 8.1 | 5.1 | 588 | 225 | SMBO |
| MnasNet-92 (Tan et al., 2019) | 25.2 | 8.0 | 4.4 | 388 | - | RL |
| DARTS (2nd order) (Liu et al., 2019) | 26.7 | 8.7 | 4.7 | 574 | 4.0 | gradient-based |
| SNAS (mild) (Xie et al., 2019) | 27.3 | 9.2 | 4.3 | 522 | 1.5 | gradient-based |
| ProxylessNAS (GPU)[‡] (Cai et al., 2019) | 24.9 | 7.5 | 7.1 | 465 | 8.3 | gradient-based |
| P-DARTS (CIFAR10) (Chen et al., 2019) | 24.4 | 7.4 | 4.9 | 557 | 0.3 | gradient-based |
| P-DARTS (CIFAR100) (Chen et al., 2019) | 24.7 | 7.5 | 5.1 | 577 | 0.3 | gradient-based |
| BayesNAS (Zhou et al., 2019) | 26.5 | 8.9 | 3.9 | - | 0.2 | gradient-based |
| PC-DARTS (CIFAR10) | 25.1 | 7.8 | 5.3 | 586 | 0.1 | gradient-based |
| PC-DARTS (ImageNet)[‡] | **24.2** | **7.3** | 5.3 | 597 | 3.8 | gradient-based |

[‡] This architecture was searched on ImageNet directly, otherwise it was searched on CIFAR10 or CIFAR100.

optimizer with a momentum of $0.9$, an initial learning rate of $0.5$ (decayed down to zero linearly), and a weight decay of $3 \times 10^{-5}$. Additional enhancements are adopted including label smoothing and an auxiliary loss tower during training. Learning rate warm-up is applied for the first 5 epochs.

Results are summarized in Table 2. Note that the architectures searched on CIFAR10 and ImageNet itself are both evaluated. For the former, it reports a top-1/5 error of 25.1%/7.8%, which significantly outperforms 26.7%/8.7% reported by DARTS. This is impressive given that our search time is much shorter. For the latter, we achieve a top-1/5 error of 24.2%/7.3%, which is the best known performance to date. In comparison, ProxylessNAS (Cai et al., 2019), another approach that directly searched on ImageNet, used almost doubled time to produce 24.9%/7.5%, which verifies that our strategy of reducing memory consumption is more efficient yet effective.

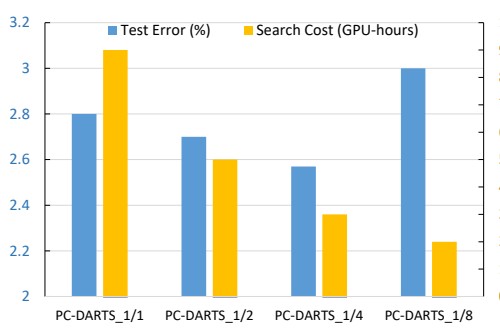

Figure 3: Search cost and accuracy comparison between our approach with different sampling rates, namely, $1/1$, $1/2$, $1/4$ and $1/8$, among which $1/4$ makes a nice tradeoff between accuracy and efficiency.

| CIFAR10 | | | |
|---|---|---|---|
| PC | EN | Test Error | Search Cost |
| ✗ | ✗ | 3.00±0.14% | 0.4 GPU-days |
| ✗ | ✓ | 2.82±0.05% | 0.4 GPU-days |
| ✓ | ✗ | 2.67±0.11% | 0.1 GPU-days |
| ✓ | ✓ | 2.57±0.07% | 0.1 GPU-days |
| ImageNet (ILSVRC2012) | | | |
| PC | EN | Test Error | Search Cost |
| ✗ | ✗ | 26.8±0.1% | 7.7 GPU-days |
| ✗ | ✓ | 26.3±0.1% | 7.7 GPU-days |
| ✓ | ✗ | 26.2±0.1% | 3.8 GPU-days |
| ✓ | ✓ | 25.8±0.1% | 3.8 GPU-days |

Table 3: Ablation study on CIFAR10 and ImageNet. PC and EN denote partial channel connections and edge normalization, respectively. All architectures on ImageNet are re-trained by 100 epochs (the 25.8% error corresponds to the best entry, 24.2%, reported in Table 2 (250 epochs).

## 4.4 ABLATION STUDY

### 4.4.1 EFFECTIVENESS OF CHANNEL PROPORTION $1/K$

We first evaluate $K$, the hyper-parameter that controls the sampling rate of channels. Note that a tradeoff exists: increasing the sampling rate (*i.e.*, using a smaller $K$) allows more accurate infor-

Table 4: Experiments on stability of DARTS and PC-DARTS. Left: Evaluations of searched architectures in five independent search runs. Middle: architectures searched with different numbers of epochs. Right: runs on architectures searched with different numbers of nodes.

| Methods | Runs | | | | | Epochs | | | | Nodes | | |
|---|---|---|---|---|---|---|---|---|---|---|---|---|
| | #1 | #2 | #3 | #4 | #5 | 50 | 75 | 100 | 125 | 5 | 6 | 7 |
| DARTS-v1(%) | 2.89 | 3.15 | 2.99 | 3.07 | 3.27 | 2.98 | 2.87 | 3.32 | 3.08 | 3.03 | 2.98 | 2.89 |
| DARTS-v2(%) | 3.11 | 2.68 | 2.77 | 3.14 | 3.06 | 2.76 | 2.93 | 3.51 | 3.18 | 2.82 | 2.76 | 3.02 |
| PC-DARTS(%) | 2.72 | 2.67 | 2.57 | 2.75 | 2.64 | 2.57 | 2.67 | 2.69 | 2.75 | 2.63 | 2.57 | 2.64 |

mation to be propagated, while sampling a smaller portion of channels casts heavier regularization and may alleviate over-fitting. To study its impacts, we evaluate the performance produced by four sampling rates, namely $1/1$, $1/2$, $1/4$ and $1/8$, on CIFAR10, and plot the results into a diagram of search time and accuracy in Figure 3. One can observe that a sampling rate of $1/4$ yields superior performance over $1/2$ and $1/1$ in terms of both time and accruacy. Using $1/8$, while being able to further reduce search time, causes a dramatic accuracy drop.

These experiments not only justify the tradeoff between accuracy and efficiency of architecture search, but also reveal the redundancy of super-network optimization in the context of NAS. More essentially, this reflects the gap between search and evaluation, *i.e.*, a better optimized super-network does not guarantee a better searched architecture – in other words, differentiable NAS approaches are easily to over-fit on the super-network. From this viewpoint, channel sampling plays the role of regularization, which shrinks the gap between search and evaluation.

### 4.4.2 CONTRIBUTIONS OF DIFFERENT COMPONENTS OF PC-DARTS

Next, we evaluate the contributions made by two components of PC-DARTS, namely, partial channel connections and edge normalization. The results are summarized in Table 3. It is clear that edge normalization brings the effect of regularization even when the channels are fully-connected. Being a component with very few extra costs, it can be freely applied to a wide range of approaches involving edge selection. In addition, edge normalization cooperates well with partial channel connections to provide further improvement. Without edge normalization, our approach can suffer low stability in both the number of network parameters and accuracy. On CIFAR10, we run search without edge normalization for several times, and the testing error ranges from $2.54\%$ to $3.01\%$. On the other hand, with edge normalization, the maximal difference among five runs does not exceed $0.15\%$. Therefore, we justify our motivation in designing edge normalization (see Section 3.3), *i.e.*, it can be a standalone method for stabilizing architecture search, yet it works particularly well under partial channel connection, since the latter introduces randomness and stabilization indeed helps.

### 4.4.3 STABILITY OF OUR APPROACH

In this part, we demonstrate the stability of our approach from three different perspectives. Results are summarized in Table 4, with detailed analysis below.

**First**, we evaluate the stability of different approaches by conducting 5 independent search runs. We re-implement DARTS-v1 and DARTS-v2 with the proposed code, as well as that of our approach, and perform five individual search processes with the same hyper-parameters but different random seeds (0, 1, 2, 3, 4). The architectures found by DARTS in different runs, either v1 or v2, suffer much higher standard deviations than that of our approach (DARTS-v1: $\pm0.15\%$, DARTS-v2: $\pm0.21\%$, PC-DARTS: $\pm0.07\%$).

**Second**, we study how the search algorithm is robust to hyper-parameters, *e.g.*, the length of the search stage. We try different numbers of epochs, from 50 to 125, and observe how it impacts the performance of searched architectures. Again, we find that both DARTS-v1 and DARTS-v2 are less robust to this change.

**Third**, we go one step further by enlarging the search space, allowing a larger number of nodes to appear in each cell – the original DARTS-based space has 6 nodes, and here we allow 5, 6 and 7 nodes. From 5 to 6 nodes, the performance of all three algorithms goes up, while from 6 to 7 nodes, DARTS-v2 suffers a significant accuracy drop, while PC-DARTS mostly preserves it performance.

Table 5: Detection results, in terms of average precisions, on the MS-COCO dataset (test-dev 2015).

| Network | Input Size | Backbone | $\times+$ | AP | $AP_{50}$ | $AP_{75}$ | $AP_S$ | $AP_M$ | $AP_L$ |
|---|---|---|---|---|---|---|---|---|---|
| SSD300 (Liu et al., 2016) | 300×300 | VGG-16 | 35.2B | 23.2 | 41.2 | 23.4 | 5.3 | 23.2 | 39.6 |
| SSD512 (Liu et al., 2016) | 512×512 | VGG-16 | 99.5B | 26.8 | 46.5 | 27.8 | 9.0 | 28.9 | 41.9 |
| YOLOV2 (Redmon & Farhadi, 2017) | 416×416 | Darknet-19 | 17.5B | 21.6 | 44.0 | 19.2 | 5.0 | 22.4 | 35.5 |
| Pelee (Wang et al., 2018) | 304×304 | PeleeNet | 1.3B | 22.4 | 38.3 | 22.9 | - | - | - |
| SSDLiteV1 (Howard et al., 2017) | 320×320 | MobileNetV1 | 1.3B | 22.2 | - | - | - | - | - |
| SSDLiteV2 (Sandler et al., 2018) | 320×320 | MobileNetV2 | 0.8B | 22.1 | - | - | - | - | - |
| SSDLiteV3 (Tan et al., 2019) | 320×320 | MnasNet-A1 | 0.8B | 23.0 | - | - | 3.8 | 21.7 | 42.0 |
| PC-DARTS with SSD | 320×320 | PC-DARTS[‡] | 1.2B | 28.9 | 46.9 | 30.0 | 7.9 | 32.0 | 48.3 |

[‡] The backbone architecture of PC-DARTS was searched on ImageNet (with a 24.2% top-1 error).

As a side note, all these algorithms fail to gain accuracy in enlarged search spaces, because CIFAR10 is relatively simple and the performance of searched architectures seems to saturate.

With all the above experiments, we can conclude that PC-DARTS is indeed more robust than DARTS in different scenarios of evaluation. This largely owes to the regularization mechanism introduced by PC-DARTS, which (i) forces it to adjust to dynamic architectures, and (ii) avoids the large pruning gap after search, brought by the *none* operator.

### 4.5 TRANSFERRING TO OBJECT DETECTION

To further validate the performance of the architecture found by PC-DARTS, we use it as the backbone for object detection. We plug the architecture found on ImageNet, as shown in Figure 2, into a popular object detection framework named Single-Shot Detectors (SSD) (Liu et al., 2016). We train the entire model on the MS-COCO (Lin et al., 2014) trainval dataset, which is obtained by a standard pipeline that excludes 5K images from the val set, merges the rest data into the 80K train set and evaluates it on the test-dev 2015 set.

Results are summarized in Table 5. Results for SSD, YOLO and MobileNets are from (Tan et al., 2019). With the backbone searched by PC-DARTS, we need only 1.2B FLOPs to achieve an AP of 28.9%, which is 5.7% higher than SSD300 (but with 29× fewer FLOPs), or 2.1% higher than SSD512 (but with 83× fewer FLOPs). Compared to the 'Lite' versions of SSD, our result enjoys significant advantages in AP, surpassing the most powerful one (SSDLiteV3) by an AP of 6.9%. All these results suggest that the advantages obtained by PC-DARTS on image classification can transfer well to object detection, a more challenging task, and we believe these architectures would benefit even more application scenarios.

## 5 CONCLUSIONS

In this paper, we proposed a simple and effective approach named partially-connected differentiable architecture search (PC-DARTS). The core idea is to randomly sample a proportion of channels for operation search, so that the framework is more memory efficient and, consequently, a larger batch size can be used for higher stability. Additional contribution to search stability is made by edge normalization, a light-weighted module that requires merely no extra computation. Our approach can accomplish a complete search within 0.1 GPU-days on CIFAR10, or 3.8 GPU-days on ImageNet, and report state-of-the-art classification accuracy in particular on ImageNet.

This research delivers two important messages that are important for future research. First, differentiable architecture search seems to suffer even more significant instability compared to conventional neural network training, and so it can largely benefit from both (i) **regularization** and (ii) **a larger batch size**. This work shows an efficient way to incorporate these two factors in a single pipeline, yet we believe there exist other (possibly more essential) solutions for this purpose. Second, going one step further, our work reveals the redundancy of super-network optimization in NAS, and experiments reveal a gap between improving super-network optimization and finding a better architecture, and regularization plays an efficient role in shrinking the gap. We believe these insights can inspire researchers in this field, and we will also follow this path towards designing stabilized yet efficient algorithms for differentiable architecture search.

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
