# OpenReview forum: "PC-DARTS: Partial Channel Connections for Memory-Efficient Architecture Search"
_ICLR.cc/2020/Conference — Accept (Spotlight)_

### Official Review · AnonReviewer1 · 2019-10-22
**Official Blind Review #1**

**Rating:** 8

**Review:**

The authors propose with this paper a simple extension of DARTS, a popular neural architecture search (NAS) method. This extension addresses one of the shortcoming of DARTS: the immense memory cost. This achieved in a simple way. Instead of using all channels only a random subset is used. To account for that, the authors propose a method to normalizes edges.

The description of the method is very clear. The related work covers many works. I suggest to focus more on the more recent work on NAS and particular work that follows the core idea of DARTS. This deserves more than used two very short sentences since this is the most related work. The experimental section leaves no question unanswered. The setup is clearly described in all cases, ablation studies are conducted wherever it is needed. The method is evaluated on both CIFAR-10 and ImageNet and is even transferred to MS COCO. The search results show some improvements with respect to time budget.
Concluding, this might not be a ground-breaking paper but it is well made and I see no obvious flaws so I do not see any reason to reject it.

**Experience Assessment:**

I have published in this field for several years.

**Review Assessment: Checking Correctness Of Derivations And Theory:**

I carefully checked the derivations and theory.

**Review Assessment: Checking Correctness Of Experiments:**

I carefully checked the experiments.

**Review Assessment: Thoroughness In Paper Reading:**

I read the paper thoroughly.

---

> ### Author Response · Authors · 2019-11-08
> **Responses to Reviewer #1**
>
> We thank the reviewer for the positive comments.
>
> Indeed, our work introduced a simple approach, partial channel connection, to regularize the architecture search process as well as create two additional benefits: (i) reduce the computational overhead to accelerate the search stage; and (ii) reduce the memory consumption so that the algorithm can be trained under a large batch size to improve search stability. In addition, we design edge normalization to compensate for the randomness caused by partial channel connection. The overall framework achieves the state-of-the-art search accuracy and speed among all DARTS-based approaches.
>
> In the related work section of the updated paper, we have used an individual paragraph to introduce DARTS and two of its variants, namely, ProxylessNAS [1] and P-DARTS [2].
>
> ===References===
>
> [1] H. Cai, L. Zhu, and S. Han. ProxylessNAS: Direct neural architecture search on target task and hardware. arXiv preprint arXiv:1812.00332, 2018.
> [2] X. Chen, L. Xie, J. Wu, and Q. Tian. Progressive differentiable architecture search: Bridging
> the depth gap between search and evaluation. arXiv preprint arXiv:1904.12760, 2019.

---

### Official Review · AnonReviewer2 · 2019-10-23
**Official Blind Review #2**

**Rating:** 6

**Review:**

** Summary **
This paper proposes to improve the previously work DARTS in terms of the training efficiency, from the large memory and computing overheads. The authors propose a partially-connected DARTS (PC-DARTS) with two components: 1. Partial channel connection 2. edge normalization. To be detailed, they sample a small part of channels to perform connection and add edge normalization to eliminate the potential optimization problem. The results on CIFAR-10 and IamgeNet show the approach is effective, especially in ImageNet, the approach achieves SOTA results.

** Strengths **
1.	The research direction of reducing the training/memory effort of Neural Architecture Search is important, which is also very hot in nowadays.
2.	The authors propose two components to perform the efficient search process, which are partial channel connection and edge normalization operations. These methods are reasonable to reduce the training effort. The authors are inspired by ShuffleNet or related research topics.
3.	The approach is easy to follow and implement, the description of the method is also clear.
4.	The experiments also show comparable results in CIFAR-10 and strong performance in ImageNet.

** Weaknesses **
1.	The operation of partial channel connection choses 1/k channels, the remain channels are directly added to the output. This operation somehow feels too straight to be reasonable, since the remain channel has larger weights (1.0) compared to previous weighted combination. Though the second edge normalization can eliminate little, but this modification still suffers from less careful design, for example, another \alpha weighted combination? Besides, directly bypass is same as perform identity operation, is this right?
2.	The motivation of edge normalization is somehow weak, as the authors are aware of this can also be applied to the original DARTS. From the ablation study, it also shows it works for the original DARTS, which makes the description of 3.3. to be not so convincing. Besides, in the first paragraph of 3.3, what does it mean that “weight-free operations often accumulate larger weights” compared to other operations? I feel the reason is that the weight-free operations are much easier to pass the gradients and easy to be trained.
3.	In imageNet results, it seems P-DARTS significantly outperform PC-DARTS in terms of the search cost, and the accuracy is similar. This makes PC-DARTS approach to be embarrassing.
4.	One general point is what the authors mentioned, indeed, for NAS, more training data involved in the training process is much more important compared to perform operations. Therefore, the advantage benefits from the less memory usage of 1/k selection and the more data in one mini-batch. This makes the design of current research directions to be different. Does it mean more training (longer time) and more GPU memory will significantly outperform current results? Even the SOTA approaches.
5.	Minor point: compared to ProxylessNAS which only samples two paths at each time, their method is much more efficient (though the binarization consumes much). What’s the most advantage of PC-DARTS compared to their method? What if combine their approach with edge normalization?


**Experience Assessment:**

I have published one or two papers in this area.

**Review Assessment: Checking Correctness Of Derivations And Theory:**

I assessed the sensibility of the derivations and theory.

**Review Assessment: Checking Correctness Of Experiments:**

I assessed the sensibility of the experiments.

**Review Assessment: Thoroughness In Paper Reading:**

I read the paper thoroughly.

---

> ### Author Response · Authors · 2019-11-08
> **Responses to Reviewer #2**
>
> We thank the reviewer for the detailed comments. Below, we provide detailed responses to each point in the weaknesses part.
>
> 1. We are not sure if we correctly understand this question. If not, please give us feedback and we will respond. Yes, directly bypass is equivalent to setting the weight of skip-connect to be 1. For the sampled channels, the total weight of all operators is 1, the same as the remaining channels. Our approach can be understood as adding a random mask on the channels so that some channels undergo mix-operator computation while others do not. The effect is regularization, which prevents the search process to overfit the super-network and thus cause instability on the performance of the preserved sub-network.
>
> We welcome more sophisticated designs that can improve our work, but we do not understand what the reviewer meant by "another alpha weight combination", and look forward to follow-up comments.
>
> 2. Sorry for being a bit misleading. Our logic is that edge normalization was inspired by improving the search stability when partial channel connection is present, *although it produces accuracy gain individually (on the original DARTS)*. We are not sure which part of Sec 3.3 was made less convincing by the fact that edge normalization works well on the original DARTS, and we look forward to future comments.
>
> Regarding weight-free operations, we did mean that "they are much easier to pass the gradients and easy to be trained", and "they accumulate larger weights" is the consequence. This phenomenon was also observed in P-DARTS [1].
>
> 3. P-DARTS is much faster since it was searched on CIFAR10, a reasonable proxy dataset for ImageNet. However, finding a proxy dataset is not always possible, and we argue that directly searching on the target dataset is an important ability of NAS approaches. In comparison, we did evaluate P-DARTS [1] with a direct search on ImageNet, but the performance is much lower than PC-DARTS. This comparison is a further justification for the contribution of PC-DARTS.
>
> 4. This does not seem to be a "weakness", but a take-away message of our approach. Note that PC-DARTS did not introduce additional training data, but allowed a larger batch size to be used which largely improves both accuracy and speed. It is well known in the community that a large batch size often leads to better and more stable performance, and we, of course, believe that GPUs with larger memory can help in this way. PC-DARTS provides an efficient solution under limited GPU memory.
>
> 5. We do not quite understand why ProxylessNAS [2] is more efficient (if "their method" means ProxylessNAS). The search space of ProxylessNAS is different from ours. Besides, ProxylessNAS required 4.0 and 8.3 GPU-days on CIFAR10 and ImageNet, while these numbers for PC-DARTS are only 0.1 and 3.8, respectively. ProxylessNAS required more computational overhead because the architecture is changing heavily from iteration to iteration (in comparison, the architectural change in PC-DARTS is channel-level and thus much lighter) so that it required a longer time for the search process to converge reasonably. Indeed, the motivation of PC-DARTS is different from ProxylessNAS, and we believe these two approaches claimed independent contributions to the community. Of course, we believe that edge normalization can be incorporated into ProxylessNAS, and we will try it in the future.
>
> ===References===
>
> [1] X. Chen, L. Xie, J. Wu, and Q. Tian. Progressive differentiable architecture search: Bridging
> the depth gap between search and evaluation. arXiv preprint arXiv:1904.12760, 2019.
> [2] H. Cai, L. Zhu, and S. Han. ProxylessNAS: Direct neural architecture search on target task and hardware. arXiv preprint arXiv:1812.00332, 2018.

---

### Official Review · AnonReviewer3 · 2019-10-23
**Official Blind Review #3**

**Rating:** 6

**Review:**

---
revised score. rebuttal clears my concerns.
---
Summary:

The paper proposes a partially connected differential architecture search (PC-DARTS) technique, that uses a variant of channel dropout for each node's output feature maps, and a weighted summation of concatenating all previous nodes. Searched architecture on CIFAR-10 and ImageNet seems to outperform the one discovered by the original DARTS, however, the results are not directly comparable due to the slight change of search space.

Introducing this edge normalization is a novel contribution, but it is more like a trick to have a better search space rather than the PC-DARTS itself. My main concerns are about the incremental novelty and experiments are heavily done on one search run, especially the search space is not the same as baseline DARTS.

I do not think the current version is ready for ICLR, but I am looking forward to seeing the authors' rebuttal and I am willing to revise my review accordingly.

Main concerns

- Incremental novelty about channel sampling.
Doing edge normalization in the PC-DARTS is indeed novel, however, the channel sampling (abbr. PC for partial channel connection) is not. Dropout is widely adopted in all deep learning training since AlexNet. In NAS with parameter sharing, Pham et al. already exploit the channel dropout as shown in ENAS function "def drop_path"(https://github.com/melodyguan/enas/blob/master/src/cifar10/image_ops.py). It is true that previous works treated like one hyper-parameters and do not provide deeper insight about this term, but it is not correct to say in Section 3.4 "Channel sampling ... has never been studied in prior work". In my perspective, the key difference of channel sampling is the retained channel number is always fixed to K, the non-selected channels are not zeroed, where the dropout usually has only a probability K / total_channel and non-selected feature is multiplied to a zero constant.

Thus, I suggest authors provide additional experiments as in Table 3 to compare the original drop-path with proposed channel sampling. Considering the test error drops from 3% to 2.67% while using PC, it will be more convincing to show the original drop path with probability K / total_channel yields a smaller drop to evidence the effectiveness of proposed sampling.

- Proposed edge normalization is not a new sampling policy but a new search space.
To my understanding, this edge normalization is effectively a change to the search space rather than the sampling policy, and generalize to many other policies as well, and can be a substantial contribution to the NAS community. However, under current experiments setting, it is hard to isolate the improvement is from this new space or the channel sampling, as detailed later.

- About the motivation.
Throughout the paper, in abstract, introduction, section 3.2 and section 4.4, the authors claim that the larger batch size is particularly important for the stability of architecture search which is not well-studied and lack of references. From Table 4, it is hard to tell the stability is from the larger batch size or the proposed partial channel sampling.


- Questions about experiments

1. Experiments comparing to the baseline is not fair.
As in Section 4.2, the CIFAR-10 search is different from the original DARTS and P-DARTS in the following manner. The batch size is changed from 64(in DARTS)/96(in P-DARTS) to 256,  super-net is freezed for the first 15 epochs, and introducing the edge normalization parameter \beta_{i,j} increase the search space.  With all these changes, it is quite hard to isolate the effectiveness of proposed PC-DARTS. Two possible simple experiments to compare is, using the original DARTS space and training set, 1) do not update the \beta but use a fixed initialization that all \beta is the same (to mimic original DARTS concatenation); 2) add \beta to original DARTS as well and re-run 1).

It is completely reasonable to me the contribution of this paper is introducing a novel edge-normalization that is simple and effective to improve the DARTS based approach. If so, the authors could revise the conclusion easily. However, in the least scenario, the experiment comparison should be in a fair way.

2. In original DARTS, error drop from 3% for the first-order gradient to 2.76% while using the second-order one, will this trend occurs with PC-DARTS?

3. Robustness
Recent work about evaluating neural architecture search reveals that NAS algorithms are sensitive to random initialization[1,2] and the search space [3], this in general leads to a notorious reproducibility problem of current NAS and shows it is not reasonable to only compare final performances on proxy tasks over **one** searched architecture. However, in the stability study in Section 4.4.3, multiple runs are still over the same architecture discovered in earlier experiments. In Section 4.4.2, the paper mentioned the search runs multiple times, yet the reported results in Table 3 are against the single run, as indicated by CIFAR-10 no PC- no EN error 3.00 +- 0.14, which is identical to the results in Table 1 DARTS (1st-order). Could the authors report the results with at least 3 different initializations, and possibly release the seeds? It would significantly strengthen the effectiveness of the proposed approach.

Minor comments

- According to Section 4.4.1 and Figure 3, change the K from 1 to 8, the search cost drops significantly. Does this mean the batch size in the ablation study is changing all the time? How could we know if the test-error is reduced due to the sampling ratio or to the batch size?

Typos
1. Table 2, caption below the table, \dag is not aligned with the one in the used table.

--- reference ---
[1] Li and Talwalker, Random search and reproducibility of neural architecture search, UAI’19
[2] Sciuto et al., Evaluating the search phase of neural architecture search, arxiv’19
[3] Radosavovic et al., On Network Design Spaces for Visual Recognition, ICCV'19.



**Experience Assessment:**

I have published one or two papers in this area.

**Review Assessment: Checking Correctness Of Derivations And Theory:**

N/A

**Review Assessment: Checking Correctness Of Experiments:**

I carefully checked the experiments.

**Review Assessment: Thoroughness In Paper Reading:**

I read the paper thoroughly.

---

> ### Author Response · Authors · 2019-11-08
> **Responses to Reviewer #3**
>
> We thank the reviewer for the detailed comments. Below, we provide detailed responses to each concern and question.
>
> [C: incremental novelty] We admit that Dropout and DropPath are widely used in this field. Partial channel connection is closely related to these methods, but the motivation behind it is quite different. Our goal, besides regularizing super-network training (same as Dropout and DropPath), also includes reducing computational overhead in both time and space, which cannot be achieved by either Dropout or DropPath. We will tune down our statement by saying "channel sampling has not been studied in NAS for reducing computational overhead". In addition, the ability to save computation comes from fixing the number of the sampled channels, so this difference is minor but important.
>
> Regarding using DropPath, we agree it is a nice diagnostic study, but since it cannot reduce memory as our channel sub-sampling method, we cannot compare it to our approach in a completely fair environment. Under a smaller batch size (as in original DARTS), DropPath and partial channel connection report comparable performance, but the former is slower.
>
> [C: edge normalization is not a new sampling policy but a new search space] This is a major misunderstanding. The search space of PC-DARTS is *identical* to that of DARTS (and other DARTS-based methods). Note that beta is a parameter to control edge selection: it stabilizes the search stage but does affect the search space. All network architectures found by PC-DARTS can also be found by DARTS. We welcome further questions of the reviewer and hope that our explanation can prevent this misunderstanding.
>
> [C: about the motivation] We did not see any prior approaches studying the impact of the batch size, but most differentiable search methods including DARTS and P-DARTS [1] tried to maximize the batch size. Also, in the field of image classification, it is well known that a larger batch size often leads to a more stable training process. The stability does come from a large batch. With K=4, we use the original batch size (64) in DARTS, and the test error (over 5 search trials) is 2.71±0.13%. Although the best model (2.60%) is comparable to that using 256-sized batches (2.57%), the worst model (2.85%) is much worse, and the average is worse and the standard deviation is larger (2.71±0.13% vs. 2.67±0.07%).
>
> [Q1: comparison is not fair] Regarding the search space issue, please refer to the above concern. For batch size and warmup training, they were also used in all our experiments of DARTS and P-DARTS [1] (except for the numbers copied from their papers). We also evaluated PC-DARTS with smaller batch sizes and obtained similar performance (2.60±0.11%) on CIFAR10, but the search time is ~2.5x longer. Note that DARTS becomes even less unstable without a warmup, meanwhile, both P-DARTS [1] and Auto-Deeplab [2] were equipped with a warmup, as claimed in the original paper.
>
> [Q2: using 2nd-order DARTS?] We tried the 2nd-order DARTS but the improvement is marginal, though being much slower. Actually, many researchers in this field pointed out that the same issue and this was the same reason why 2nd-order DARTS was not used in these works, including P-DARTS [1], ASAP [3] and ProbNAS [4].
>
> [Q3: robustness?] Sorry for incorrect statements in the paper. Indeed, all the standard deviation numbers reported in Tables 3 and 4 are recorded on *independent search and evaluation*, not using the same searched architecture. We have fixed this claim in the updated paper. The seeds we have used for five individual runs are (0, 1, 2, 3, 4). Thanks for correcting this point!
>
> [Minor: impacts of batch size?] Yes, the batch sizes in the experiments of Sec 4.4.1 changes all the time (always proportional to K, so that we make full use of GPU memory). We do this for acceleration, but changing batch size does not impact search accuracy a lot. To verify this, we used a batch size of 64 for K=4 (same as K=1) and the test error is 2.60%, comparable to 2.57% reported in the paper.
>
> [Typos] We have fixed them in the updated paper. Thanks!
>
> ===References===
>
> [1] X. Chen, et al., Progressive differentiable architecture search: Bridging
> the depth gap between search and evaluation. arXiv preprint arXiv:1904.12760, 2019.
> [2] C. Liu, et al., Auto-DeepLab: Hierarchical neural architecture search for semantic image segmentation. arXiv preprint arXiv:1901.02985, 2019.
> [3] A. Noy, et al., ASAP: Architecture Search, Anneal and Prune. arXiv preprint arXiv:1904.04123, 2019.
> [4] F. P. Casale, et al., Probabilistic Neural Architecture Search. arXiv preprint arXiv:1902.05116, 2019.

---

> > ### Comment · AnonReviewer3 · 2019-11-15
> > **reviewer response**
> >
> > Thanks for providing the responses to all of my questions, and sorry for the late reply. Please update the main text (as much as possible, I understand this is a short period of time) accordingly and I will make a pass again after this discussion period to provide my final recommendation.
> >
> > My response to your rebuttal:
> >
> > About edge normalization, I understand your point that these additional architecture parameters are not used during extracting the search phase. Nevertheless, this idea itself is novel and should be valued.
> >
> > Thanks for the DropPath comment, I see your approach is a different way (easier to be implemented to be efficient computation). It is a contribution, but I suggest the authors tune down and acknowledge the previous works. Otherwise, people new to this field may get the wrong impression that this is the first work to do such channel dropping.
> >
> > For robustness, please cite the works I mentioned in the updated version. In fact, your approach seems to constantly out-performs DARTS hence outperforms the random search baselines in [1,2], which is a great step in weight sharing NAS.
> >
> > **Additional question on your reported values**
> > Table 1: best result of PC-DARTS = 2.57 +- 0.07.
> > Table 3: with PC/EN, the same as Table 1, but you said it is an average across multiple runs.
> >
> > It contradicts to your Table 4, where I compute the average PC-DARTS over 5 runs ~= 2.67 (best is 2.57).
> >
> > The same problem goes for DARTS-v1 results.
> >
> > Please correct these contradicting statistics in the final version, or provide reasonable explanations. These mismatches drastically undermine the credibility of your experiments.
> >
> >
> > --- reference ---
> > [1] Li and Talwalker, Random search and reproducibility of neural architecture search, UAI’19
> > [2] Sciuto et al., Evaluating the search phase of neural architecture search, arxiv’19

---

> > > ### Author Response · Authors · 2019-11-15
> > > **Thanks for your response**
> > >
> > >
> > > We thank the reviewer for providing us with these valuable responses! We are glad to see that the reviewer agreed with our points that (1) edge normalization does not introduce a new search space, and (2) our approach of channel sampling makes some difference which helps to improve computational efficiency. We hope that major concerns have been addressed by the rebuttal.
> > >
> > > We are trying our best to update the paper, but, given that the deadline gets very close, we are not sure that everything will be done perfectly. Of course, we will do everything we can to provide an updated preprint for discussion and guarantee the quality of this paper in the final version.
> > >
> > > Regarding the new comments, below are our responses.
> > >
> > > Q1: I suggest the authors tune down and acknowledge the previous works. Otherwise, people new to this field may get the wrong impression that this is the first work to do such channel dropping.
> > >
> > > A1: Thanks, we have explained the difference between channel sampling and DropPath in the current version.
> > >
> > > Q2: For robustness, please cite the works I mentioned in the updated version.
> > >
> > > A2: Thanks. We have cited them in the current version.
> > >
> > > Q3: Additional question on your reported values: some mismatches.
> > >
> > > A3: Thanks for this question. We followed the convention used in both DARTS and P-DARTS. We searched using DARTS-v1, DARTS-v2, and PC-DARTS for 5 times. The detailed numbers are recorded in the first part of Table 4. One can see that the *average* error rate over 5 runs is 2.67%, and the *best* one is 2.57%.
> > >
> > > We used a standalone validation process to pick the best one from the 5 runs. This process was done by using 45,000 out of 50,000 training images for training, and the remaining 5,000 for validation. The one that reported the best result in the validation set was used in the *real* testing stage, which reported a test error of 2.57% and appeared in Table 1. By the way, 2.57% itself is an average over 3 individual runs (re-training) on the *same* architecture, and +-0.07 is the standard deviation of these 3 runs.
> > >
> > > So, we decide to keep the error rate we reported in these tables. Of course, we have added elaboration under Table 1 in the current paper to avoid misunderstanding. Thanks again!

---

### Decision · Program_Chairs · 2019-12-19

**Decision:**

Accept (Spotlight)

**Comment:**

This paper proposes an improvement to the popular DARTS approach, speeding it up by performing the search in a subset of channels. The improvements are robust, and code is available for reproducibility.

The rebuttal cleared up initial concerns, and after the (private) discussion among reviewers now all reviewers give accepting scores. Because the improvements seem somewhat incremental and only applied to DARTS, R3 argued against an oral, and even the most positive reviewer agreed that a poster format would be best for presentation.

I therefore strongly recommend recommendation, as a poster.